# Creating Collaborative Augmented Reality Experiences for Industry 4.0 Training and Assistance Applications: Performance Evaluation in the Shipyard of the Future †

**Aida Vidal-Balea** [1,2,*], **Oscar Blanco-Novoa** [1,2], **Paula Fraga-Lamas** [1,2,*], **Miguel Vilar-Montesinos** [3] **and Tiago M. Fernández-Caramés** [1,2]

[1] Department of Computer Engineering, Faculty of Computer Science, Universidade da Coruña, 15071 A Coruña, Spain; o.blanco@udc.es (O.B.-N.); tiago.fernandez@udc.es (T.M.F.-C.)
[2] Centro de Investigación CITIC, Universidade da Coruña, 15071 A Coruña, Spain
[3] Navantia S. A., Astillero de Ferrol, 15403 Ferrol, Spain; mvilar@navantia.es
[*] Correspondence: aida.vidal@udc.es (A.V.-B.); paula.fraga@udc.es (P.F.-L.); Tel.: +34-981167000 (P.F.-L.)
[†] This article is an extended version of a paper titled "A Collaborative Augmented Reality Application for Training and Assistance during Shipbuilding Assembly Processes", which was presented at the 3rd XoveTIC Conference, A Coruña, Spain, 8–9 October 2020.



**Featured Application: The proposed collaborative Industrial Augmented Reality (IAR) framework enables creating IAR experiences easily to facilitate, support, and optimize production and assembly tasks through training and assistance. The performance of such a framework is evaluated in terms of packet communications delay, communication interference, and anchor transmission latency. The proposed system is validated in a shipyard, thus providing useful insights and guidelines for future developers of Industry 4.0 applications.**

**Abstract:** Industrial Augmented Reality (IAR) is one of the key technologies pointed out by the Industry 4.0 paradigm as a tool for improving industrial processes and for maximizing worker efficiency. Training and assistance are two of the most popular IAR-enabled applications, since they may significantly facilitate, support, and optimize production and assembly tasks in industrial environments. This article presents an IAR collaborative application developed jointly by Navantia, one of the biggest European shipbuilders, and the University of A Coruña (Spain). The analysis, design, and implementation of such an IAR application are described thoroughly so as to enable future developers to create similar IAR applications. The IAR application is based on the Microsoft HoloLens smart glasses and is able to assist and to guide shipyard operators during their training and in assembly tasks. The proposed IAR application embeds a novel collaborative protocol that allows operators to visualize and interact in a synchronized way with the same virtual content. Thus, all operators that share an IAR experience see each virtual object positioned at the same physical spot and in the same state. The collaborative application is first evaluated and optimized in terms of packet communications delay and anchor transmission latency, and then, its validation in a shipyard workshop by Navantia's operators is presented. The performance results show fast response times for regular packets (less than 5 ms), low interference rates in the 5 GHz band, and an anchor transmission latency of up to 30 s. Regarding the validation tests, they allow for obtaining useful insights and feedback from the industrial operators, as well as clear guidelines that will help future developers to face the challenges that will arise when creating the next generation of IAR applications.

**Keywords:** Industry 4.0; augmented reality; industrial augmented reality; assistance; training; collaborative applications; Microsoft HoloLens

---

## 1. Introduction

The Fourth Industrial Revolution, also called the Industry 4.0 paradigm, represents the digital transformation of factories in terms of people, processes, services, and systems to increase competitiveness and offer a customer-centered value. The term was originally coined by the German government in 2011 [1,2]. One of the foundations of Industry 4.0 consists of gathering as much data as possible from the value chain and acquiring intelligence to drive smart manufacturing. Another aim is to achieve the convergence between the physical and digital world through solutions like cyber-physical systems or digital twins [3]. A digital twin can be defined as a virtual representation of a physical asset enabled through data and simulations for real-time monitoring and actuation during operation [4].

With the objective of providing such a level of interconnection, smart factories rely on communications architectures that enable information exchange among multiple industrial devices distributed throughout a factory, autonomous decision-making processes, and end-to-end supply chain traceability. The Industry 4.0 paradigm makes use of relevant disruptive technologies. Examples of such enabling technologies are cloud and edge computing [5], blockchain [6], artificial intelligence [7], the Industrial Internet of Things (IIoT) [8], robotics [9], and Augmented Reality (AR) [10].

Under the Industry 4.0 paradigm, shipbuilding companies are updating and integrating new technologies in their working processes. Among the enabling technologies that can be used in a so-called Shipyard 4.0 [11], one of the most promising is AR, specifically Industrial Augmented Reality (IAR), which provides a wide range of features that can be leveraged in order to develop powerful and useful applications able to show virtual elements in real scenarios.

Navantia, one of biggest European shipbuilders, created with the University of A Coruña (UDC) a Joint Research Unit Navantia-UDC to design, to implement, and to validate different Industry 4.0 technologies through various research lines. The authors of this article work in a research line called "Plant Information and Augmented Reality", devoted to analyze the potential of IAR applications for Shipyard 4.0.

AR's first developments appeared in the early 1960s with the experiments of Ivan Sutherland [12]. In the 1990s, AR gained some traction as a technology that allows for mixing the physical and digital world through natural interactions and 3D models. Recent advances in software and hardware capabilities enable developing high-quality and cost-effective AR experiences with different devices, from simple low-cost smartphones to very sophisticated Head-Mounted Display (HMD) devices. Industry 4.0 AR applications are relatively new in comparison to entertainment [13] or gamification [14] applications. This is due largely to the more restrictive requirements in tracking, registration accuracy, and end-to-end latency, as well as to the specific characteristics of industrial environments.

IAR can help to improve industrial processes, to maximize worker efficiency and mobility, and to incorporate new learning and training procedures such as blended teaching and digital near-the-job trainings [15]. Such trainings can reduce costs and time, increase safety, and adapt to different learning paces and experience. Although the advantages of IAR technologies have already been empirically validated [16–18], there are currently only some early works that apply such technologies in a collaborative and targeted manner in production environments [19].

This article presents the analysis, design, implementation, and evaluation of a smart glasses-based IAR application aimed at training and assisting Navantia's operators in assembly tasks. In contrast to our previous work [20], which was mainly focused on describing the developed application, this article focuses on the embed collaborative framework and the evaluation of its performance. Specifically,

this article provides the following main contributions, which, as of writing, have not been found together in the literature:

- The potential of IAR to facilitate, support, and optimize production and assembly tasks through training and assistance IAR applications is analyzed.
- Thorough details on the analysis, design, and implementation of an IAR application based in Microsoft HoloLens smart glasses [21] are provided so as to ease the work of future developers when developing similar IAR applications.
- A novel collaborative IAR framework is proposed in order to enable creating IAR experiences easily. The performance of such a framework is evaluated in terms of packet communications delay, communication interference, and anchor transmission latency.
- The validation of the proposed system by Navantia's operators is presented, thus providing useful insights and guidelines for future developers.

The rest of this paper is structured as follows. Section 2 reviews the latest academic and commercial IAR systems for training and assistance, enumerating the most relevant shortcomings that motivated this work. Section 3 presents the design requirements and the communications architecture of the IAR system, while Section 4 provides details on the implementation and on the proposed collaborative protocol. Section 5 describes the performed experiments and validation tests, analyzing the key findings and the future challenges. Finally, Section 6 is devoted to the conclusions.

## 2. State-of-the-Art: IAR Applications for Training and Assistance

### 2.1. IAR Training Systems

Well-trained operators have a major impact on productivity. IAR can help during training processes by providing contextual information and step-by-step instructions when performing specific tasks. Moreover, IAR applications can help after the execution of a task in order to monitor the performance of the trainee. Such support and feedback are very important when training workers to operate machinery like the one used for assembling in sequence, since they reduce the time and effort dedicated to check documentation [22] and enhance the accuracy (e.g., by reducing the error rate in assembly tasks [23]) and efficiency of the performed task. Therefore, IAR can reduce training time for new employees and lower their skill requirements by significantly decreasing the impact of previous experience in the learning process. Moreover, the displayed instructions can be adapted to the previous experience of the workers. In this regard, IAR emerges as a human-centered tool to assist non-expert and less skilled operators in executing new tasks.

There are some interesting examples of the benefits of IAR training in the literature. For instance, in [16], the authors evaluated a virtual training system for the assembly of a pneumatic cylinder through both AR and Virtual Reality (VR) applications. Based on a 16 people evaluation, the results showed significant differences in the system usability between AR and VR, but no discrepancy in terms of ergonomics and the perceived task load. As a result, some recommendations for the design of AR-based systems were indicated. Such guidelines were given in terms of task, user, information, interaction, and technology adequacy.

### 2.2. IAR Assistance Systems

Training and assistance are two of the most common IAR applications. Assistance systems aim to provide operators and supervisors with visual or auditory information to perform a task [24]. Such information is rendered ubiquitously and seamlessly, to be perceived in a context-aware manner.

In the literature, IAR has already proven to be useful in a number of previous assistance tasks like the interaction with robots or in guidance systems. For instance, the authors of [25] used AR as an enhanced user interface designed to use visual cues to improve users' spatial perception and provide more accurate dimensions and distances when guiding a telepresence robot. Other recent works have

also presented AR-based tools for controlling robot movements [26]. The interested reader can find a systematic review of AR-based remote guidance systems in [27].

With respect to common Industry 4.0 processes, an example was presented in [28], where the researchers presented a simple use case of an assembly simulator. A similar work with a smartphone-based application that uses Vuforia and Unity software was described in [29]. Such an application was aimed at training and assistance when handling industrial equipment (e.g., PLCs), and its usability was evaluated through a questionnaire.

Teleassistance is also important when monitoring and repairing machines. Thus, teleassistance systems can ease remote collaboration between workers [30] and provide the same level of information that would be shared when being physically together. Such a type of augmented communication can also be used for collaborative visualization in the different engineering processes [31]. In addition, IAR systems are also able to give quick access to documentation like manuals, 3D models, or historic data [32]. Moreover, IAR is helpful when supporting decision making in real scenarios by combining the physical experience with displayed information that is extracted in real time [33].

Recently, more sophisticated approaches have been proposed. For instance, an olfactory-based AR system to help with the identification of maintenance issues was proposed [34]. In addition, in [35], the authors aimed to enhance a simulation-based assistance application by adding contextual awareness from the sensors. Another work that included Internet of Things (IoT) interactions was proposed in [36], where a framework was described for Microsoft HoloLens smart glasses that eases the integration between AR and IoT devices. Furthermore, IAR can contribute significantly to the definition of the properties of a digital twin by modeling logical objects within a virtualized space and then associating such a logic with physical entities [37].

## 2.3. Developing IAR Training and Assistance Systems

Both training and assistance systems use IAR features to deliver information related to the task execution. Such features can be categorized mainly into two main types [38]: embodiment/awareness and virtual fixtures. Embodiment enhancement [39] comprises information that can improve the overall illusion of presence, for example with the use of an HMD. Virtual fixtures [40] are additional objects (e.g., virtual images, text, arrows) overlaid on a remote scene, that help the operator/supervisor perceive the remote environment and the different points of interest. Such virtual fixtures are also helpful for training when highlighting important information regarding the task, thus guiding the trainee step-by-step to accomplish the task and/or to give graphical feedback. For instance, the benefits of haptic and graphical feedback on surgeon performance during a teleoperated palpation task of artificial tissues were presented in [41]. In addition, in [38], the researchers studied the effects of different AR features in an assembly scenario. In particular, the paper sought to understand to what extent excessive visual information can be detrimental; and what is the effect of users' previous expertise.

There are several aspects that make both IAR assistance and training-based systems very complex, such as communication limitations like high latency or human factors like perceptual issues and visualization of the remote environment. Recent literature has studied the degradation of the functionality of the human visual system when using head-worn AR displays [42]. Some of the factors that have been considered in the literature are lighting conditions, decreased visual acuity, limited contrast, distorted perception of colors, identification of the different objects or parts, spatial perception, and the subjectively perceived distance (e.g., egocentric distance [43]). Moreover, additional consideration is given to the need for a short familiarization time with the AR system to perform assistance and training complex tasks. Regarding the performance evaluation of IAR applications, in [44], the authors studied the outcomes of the given instructions. Factors like task completion time, error rate, cognitive effort, and usability were assessed. For instance, it was also analyzed how people with previous AR/VR/gaming experience obtain better results at performing

tasks in an AR scenario since they already possess skills such as judging distances or finding adequate strategies of motion [38].

### 2.4. Shipbuilding IAR Systems

Some of the previously analyzed works share constraints with the shipbuilding sector, like construction [45,46]. Nevertheless, there are not many assistance or training systems in the literature specially devoted to shipbuilding. Most of the literature found is focused on pipes, one of the key parts that is manufactured in a shipyard. For instance, the work detailed in [47] focused on pipe management through markers. Another pipe-related work was presented in [48], where the authors visualized the design dimensions of pipes to adjust them to the actual installation dimensions. A very recent implementation was presented in [49]. The authors proposed a Microsoft HoloLens AR-assisted guidance system for an assembly task. The application acts as a data-visualization interface for files obtained from 3D-modeling and process-planning software. The feasibility of the system was evaluated through a test carried out by six students. The obtained results showed an increase of the assembly efficiency of 25.6%. Nevertheless, the results of an evaluation questionnaire indicated that although the system is user-friendly, Microsoft HoloLens devices still have room for improvement.

Regarding the different available commercial solutions, recent works have already reviewed the most relevant IAR commercial hardware and software tools and evaluated the performance of some of them in a shipyard workshop and inside a ship under construction [50]. With respect to specific training and technical assistance applications, there are also several software solutions that come either from companies like PTC (Vuforia Enterprise AR Suite), TeamViewer (TeamViewer), Microsoft (Dynamics 365), Atos (Air Assist [51]), Capgemini (Andy 3D [52]), Sonovision (Reflekt [53]), Librestream [54], Wideum (Remote Eye [55]), and Innovae (ATR [56]). The analyzed solutions have similar characteristics, but they lack flexibility and do not fully fulfill the requirements of a shipyard. This is due to the fact that they are closed proprietary solutions that depend on the developing company when adding new functionality, so they incur additional costs for paying for licensing or subscription fees. Moreover, for some requirements, it may be necessary to create an additional ad-hoc application that may not be integrated directly into the original IAR solution. In addition, the ideal IAR solution should be compatible with different devices smartphones, tablets, AR smartglasses (e.g., ODG, Vuzix, Google, Epson Moverio, Realwear), and different collaborative platforms with different operative systems. Furthermore, it must be considered that the data must be owned, controlled, and managed by the shipbuilder, since they are usually considered as critical information, especially for military vessels. Finally, the solution should be designed to be fully integrated and operative with other products and services of the shipbuilding company (e.g., digital twin, Enterprise Resource Planning (ERP), Product Lifecycle Management (PLM), Manufacturing Execution System (MES), cloud services, IoT services).

### 2.5. Analysis of the State-of-the-Art

After analyzing the previously mentioned work of the state-of-the-art, it is possible to highlight some shortcomings that motivated the creation of this work:

- First of all, there is a lack of solutions devoted specifically to solving the main issues related to AR training and assistance in shipbuilding.
- Second, there are not many solutions designed for Microsoft HoloLens, currently one of the more sophisticated HMD devices.
- Third, there is no significant recent literature that proposes a collaborative framework for IAR. There are only some early works that were systematically reviewed in [57].
- Finally, the fourth shortcoming is related to the fact that, although different experiments have been proposed mainly regarding visual fixtures and on the usability of IAR applications, there are

no significant solution validations that include performance considerations for their deployment under production conditions in a real-world industrial scenario.

This article considers the previously described issues and proposes a complete solution to tackle them.

## 3. Analysis and Design of the Proposed System

This section provides insights into the process of creating AR-enabled Industry 4.0 applications. Thus, the next subsections detail the most relevant stages of the design of the proposed system, which can be easily replicated by other developers interested in developing the next generation of IAR applications.

### 3.1. Main Goals of the System

The objective of the proposed system is to exploit the capabilities of AR to facilitate, support, and optimize production and assembly tasks in an industrial workshop. With this objective, the next sections describe the development and testing of an application based on the Microsoft HoloLens smart glasses that is able to guide the operators and facilitate their training tasks in a visual and practical way.

The developed application allows for the visualization of industrial 3D models in which the assembly steps to be carried out in each part of the process are highlighted by means of animations. The application seeks to reduce assembly times and to visually illustrate the contents of the assembly manuals, which in most shipyards are still only available for the operators on paper. Thus, the developed system is expected to optimize the processes and reduce the risks associated with errors during assembly.

Specifically, the developed application was designed having in mind the following main goals:

- It should allow for scaling and moving the displayed 3D models with the aim of facilitating their visualization, thus enabling placing them where the user considers appropriate. In addition, when moving an object, an automatic scanning of the environment will be carried out to avoid the model being placed in an intersection with real objects, as this would cause an uncomfortable effect when visualizing it.
- The assembly sequence should be visualized step-by-step through animations and contextual text instruction that provide the necessary technical details.
- It should be possible to visualize the documentation associated with every relevant part, especially its blueprints and physical measures.
- The developed application should implement a shared experience system that allows multiple AR devices to interact with the same virtual part at the same time.
- Ease of use: The application interface should be as simple and intuitive as possible to avoid misunderstandings during the training process. Animations and interactions should also be reinforced through the reproduction of sounds that guide each action (e.g., start and end of the animations, selections in the menus).

### 3.2. Design Requirements

The design requirements of the system were divided into four categories: basic functionalities, hydraulic clutch (i.e., the virtual object), documentation, and shared experiences. Each of them can be seen as a different module of the application.

The following are the requirements for the basic functionalities of the system:

- User interaction with the system through the gestures detected by the HoloLens smart glasses (gaze and tap).
- The cursor that indicates the position the user is looking at (gaze) at any given moment.

- The panel through which the user can interact with the part to scale it, move it, start animations, or read additional information about it.
- Panel tracking of the user, so that when the user changes her/his location or rotates his/her head, the panel is positioned in front of the user. A button should also be included on the panel to enable or disable this functionality.
- A modular internationalization system for real-time translation of the application, allowing more languages to be easily added.
- Sounds and sound effects that help the user recognize and reaffirm different actions, such as a click or the start and end of the animations.

Below are the requirements for the 3D model of the hydraulic clutch:

- Creation of animations that illustrate the assembly steps contained in the part assembly manual.
- Creation of an assembly-disassembly animation that makes all the parts of the clutch visible.
- Scaling of the 3D model so that it can be made bigger and smaller.
- Movement of the 3D model. During this process, the surrounding surfaces are mapped by the HoloLens to avoid possible collisions and intersections of the part with real elements.

The following are the requirements related to the documentation of each of the virtual parts:

- The parts that have associated documentation should be highlighted to facilitate their detection and interaction by the user.
- There should be an independent panel that shows the documentation available for each part.
- There should be an independent panel that shows the manufacturing order and the hydraulic clutch assembly manual.

Finally, the requirements of the shared experience module are the following:

- The AR environment should be synchronized among multiple HoloLens devices.
- Sharing of the hydraulic clutch, taking into account its location and rotation, as well as its real-time status, so that all users share their interactions with the part (animations, scaling, movement) with the rest of the used HoloLens glasses.

*3.3. Communications Architecture*

Figure 1 shows the communications architecture of the proposed system. At the bottom of the figure are two collaborative AR networks that are composed by HMD devices. Each network has a master device, which synchronizes the details of the overall AR scene, and several slave devices, which follow the indications of the Master device, but which also can communicate among them to share relevant events (e.g., an event that is triggered when a user clicks on a shared virtual object).

The devices of a collaborative AR network are usually physically close and communicate among them, but they can also communicate with devices of other remote collaborative AR networks in order to share the same AR experience. Such communications are carried out through WiFi Access Points (APs) and over the deployed wired communications infrastructure.

At the top of Figure 1 is the cloud, which in the case of Navantia runs on a cluster of PCs that execute the software that is necessary for the creation of AR content (e.g., NX or FORAN for Computer-Aided Design (CAD) model creation) or for providing contextual data to an AR scene (e.g., Production Engineering and Operations (PEO) and ERP software from SAP).

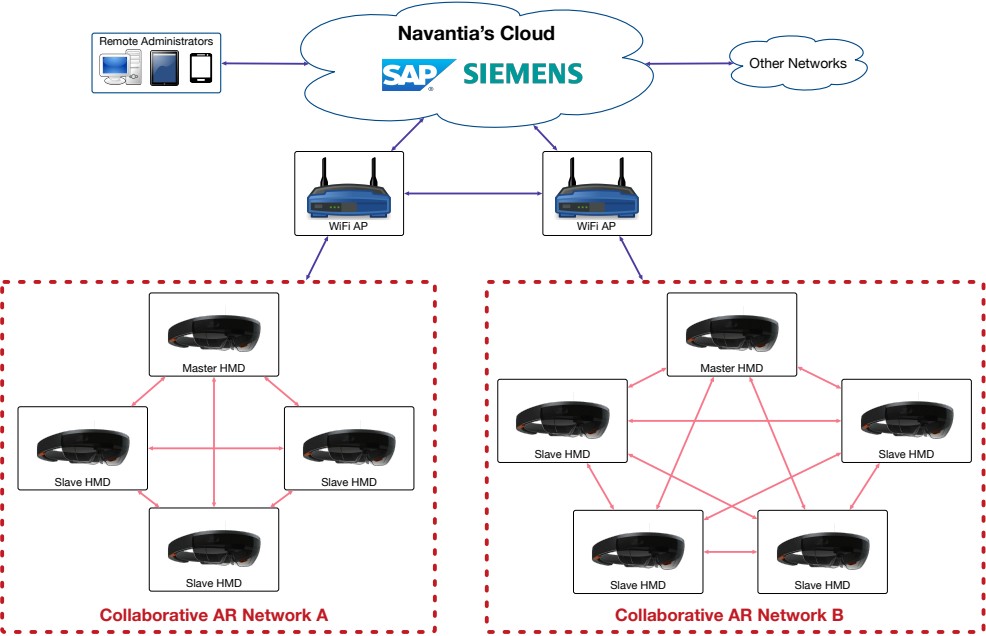

**Figure 1.** Architecture of the proposed collaborative AR system.

## 4. Implementation

### 4.1. Hardware and Software

In order to implement the proposed system, it was necessary to use different software and hardware, both for the development of the application and for its execution and visualization. Specifically, the proposed IAR application was developed using two Microsoft HoloLens first-generation smart glasses. For most of the development of the application, Unity 2019.3.3 [58] and Microsoft HoloToolkit were used to implement 3D model interaction and visualization. HoloToolkit provides certain features like gesture recognition and built-in buttons that simplify the creation of interaction events with the 3D model.

The application runs autonomously on the smart glasses. A local network connection is only necessary if several users want to share the AR experience. In such a case, the devices share the information about the state of the environment among themselves so that all the participants in the experience see the 3D elements at the same position at the same time.

### 4.2. Collaborative Framework

To synchronize the multiple HoloLens users, a collaborative framework was devised and implemented. Such a framework was based on a protocol that relies on master-slave communications where any node can take both roles depending on what the system needs. When a node (i.e., a pair of smart glasses) joins the network, it initiates a discovery process over UDP. This process allows any node to know whether there is already a master node on the network, and depending on the answer, the master or slave role is assigned to the joining node.

Different existing protocols and frameworks were taken into consideration to implement this system. A comparison of some of the systems is shown in Table 1. After studying the advantages and disadvantages of each of them, it was decided to use a custom approach based on TCP and UDP because it offers the most flexibility, as well as being robust and lightweight. Such a flexibility is really important in the context of a shipyard where the framework will be integrated with other systems in the future. Furthermore, providing all the functionality locally is more appropriate for the type of confidential (i.e., military) projects that are managed by the shipyard where the application was validated.

**Table 1.** Comparison of the different frameworks considered for the implemented system.

| Framework | Relevant Features | Limitations |
|---|---|---|
| RAKNet | Native support C++, fast, multi-platform | Complex, requires a server |
| UNET | Integrated with Unity, standard | Deprecated by unity, no replacement yet |
| Photon | Integrated with Unity, used by Microsoft (2020) | Proprietary, requires cloud license |
| Mirror | Integrated with Unity, open source, network discovery | Hard to make multi-platform |
| Custom UDP-TCP | Lightweight, simple, custom made for the needs | Not standard |

### 4.2.1. Master's Discovery Process

During the discovery process, the joining node sends HELLO messages as broadcasts and waits for the master's answer. If there is a master on the network, it sends a response via TCP to the user. This process is illustrated in Figure 2, where, as an example, the interaction among three nodes during the discovery process is depicted. Solid lines represent TCP packets, and dotted lines represent UDP packets. It should be noted that all nodes receive the HELLO message, but the slaves ignore it, while the master responds to it.

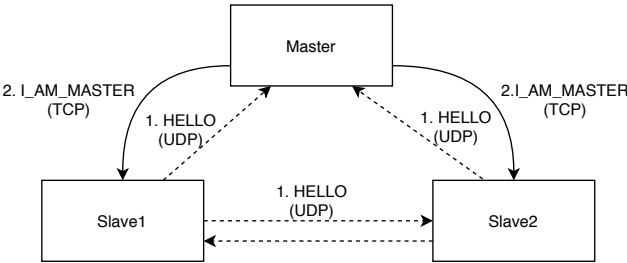

**Figure 2.** Discovery process message exchange.

### 4.2.2. Anchor Synchronization

In order to share the same AR experience, AR devices make use of anchors, which are entities used to align virtual objects in the same physical location and rotation (thus, multiple users see the virtual elements in the same physical position). In the implemented protocol, once a node receives the role of slave, it asks for the shared anchor directly from the master node over TCP (this process is illustrated in Figure 3). Since an anchor is usually sent in a large packet (often between 60 MB and up to hundreds of MBs), the use of TCP instead of UDP ensures that certain communications errors that may occur during the data transmission are automatically handled. During the synchronization process, the change of state of the anchor is indicated by different colors: red (system ready), blue (requesting anchor), yellow (importing anchor), and finally, green (anchor synchronized).

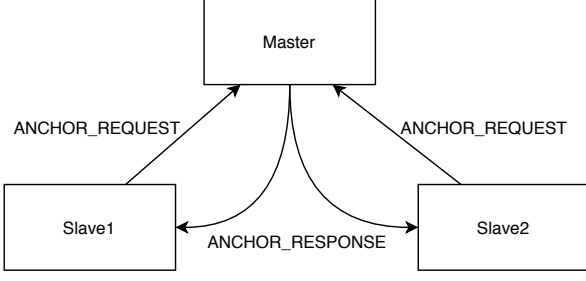

**Figure 3.** Anchor request message exchange.

### 4.2.3. AR User Event Synchronization

When all nodes are synchronized, they send their updates through small UDP packets as broadcast messages of about 30–50 bytes, so that all other nodes are aware of the changes in the shared experience.

Figure 4 shows an example of the exchanged messages when a master node broadcasts a shared "move" message, which indicates that a 3D object has changed its position. In addition, Figure 4 also shows that Slave1 sends a broadcast with a shared click message, which indicates that the user has interacted with an element of the environment.

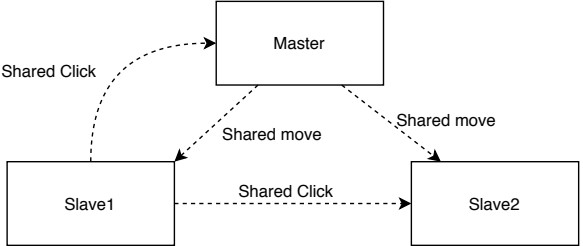

**Figure 4.** Shared messages' exchange.

Figure 5 illustrates the message exchange flow. In such a sequence diagram, the beginning of the discovery process starts once the first AR node (i.e., the first pair of HoloLens glasses) executes the application. If there are no other devices on the network, no one will respond to the HELLO messages, so the first node will assume the role of master. When a second AR device connects to the network, it performs the discovery process as indicated, but then, the master will respond; thus, the second user assumes the role of slave. Once this point is reached, the connected devices start to automatically exchange messages with the objective of keeping track of the actions carried out by the different nodes and thus synchronizing the shared experience among all the devices. When a new user joins the shared experience (like the third user shown on the right of Figure 5), the application has to update the new user state, since the previous messages sent by the other two users will not be forwarded by them.

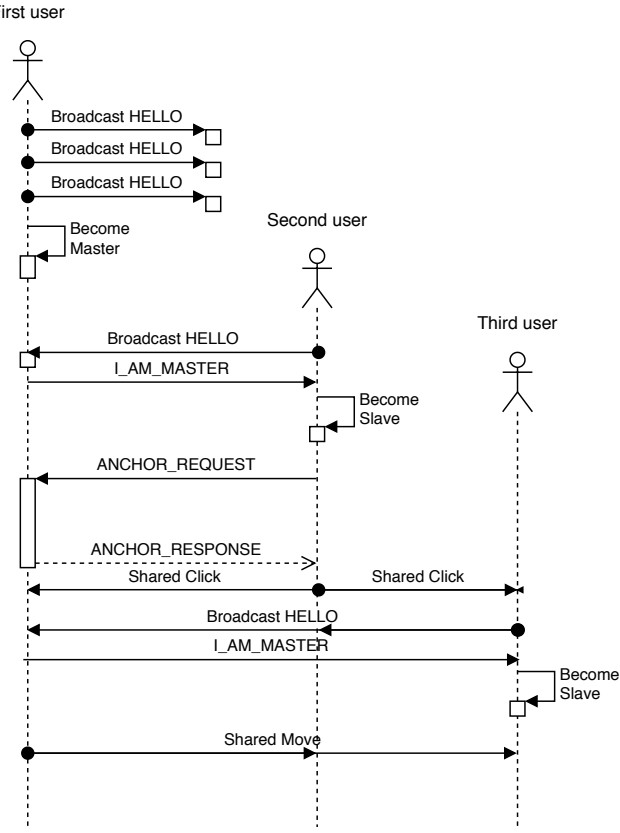

**Figure 5.** Flow diagram of a simple message exchange.

There might be situations when actions from different users are contradictory; for example, when two users try to move the same object to different directions. This is considered to be an implementation-specific problem that every application may resolve in a different way. For instance, a flag can be set to act as a semaphore and then block the movement of a specific object to other users while such an object is moved. On the contrary, other solutions may opt for allowing users to take control of objects by requesting them from other users. Therefore, the specific implementation will depend on the desired behavior of each application. In the implementation described in this article, when users try to move an object, its location is smoothed by the framework between the new position set by the local user and the new one sent over the network, so the user will experience a "spring effect", as if both users were "fighting" to move the object.

### 4.3. HoloLens Application

#### 4.3.1. Design

The implemented HoloLens application is based on the use of a 3D model of a hydraulic clutch (provided by Navantia's Turbine workshop) and on several PDF files that document the assembly process.

The first step of the development consisted of defining the structure of the application and dividing the assembly steps into different sequences that would be later animated with Unity. The structure of the Unity scene and its components is defined as:

- Camera: This is the Unity camera used to render the objects in the scene. At the beginning of the execution, it must be located in the XYZ (0,0,0) coordinates with no rotation in any of the axes.
- Directional lights: The lights that illuminate the scene in a homogeneous way.
- Manager: The empty GameObject that contains the general logic of the scene: GazeManager, GazeGestureManager (responsible for capturing gestures and the communications with the selected objects by sending an "OnSelect" message) and LocalizationManager (this component handles the movement of virtual elements).
- Cursor: The object located at the collision point between the gaze and the rest of the scene. The cursor is placed on the surfaces that can be interacted with to make it clear to the user that they can be selected.
- Panel/panels: Each panel contains buttons that are needed for interacting with the hydraulic clutch. Each button sends a message to the clutch so that the corresponding action is performed.
- Clutch/turbine: The 3D object of the clutch that is composed of all its parts.

Figure 6 shows a simplified class diagram that represents the main components of the system. Most of the logic of the system is designed to be performed through Unity commands, which allow for keeping the virtual object behavior decoupled from the rest of the IAR application.

The *ObjectMovementController* script manages the behavior of the clutch while it is moving. The clutch will always face the user. Furthermore, the clutch will be hovering at the position the user is looking. During the movement process, *UnitySpatialMappin* will be activated, so that the surface detected by the HoloLens will begin to be visible and the clutch will be prevented from being placed to collide with a real object.

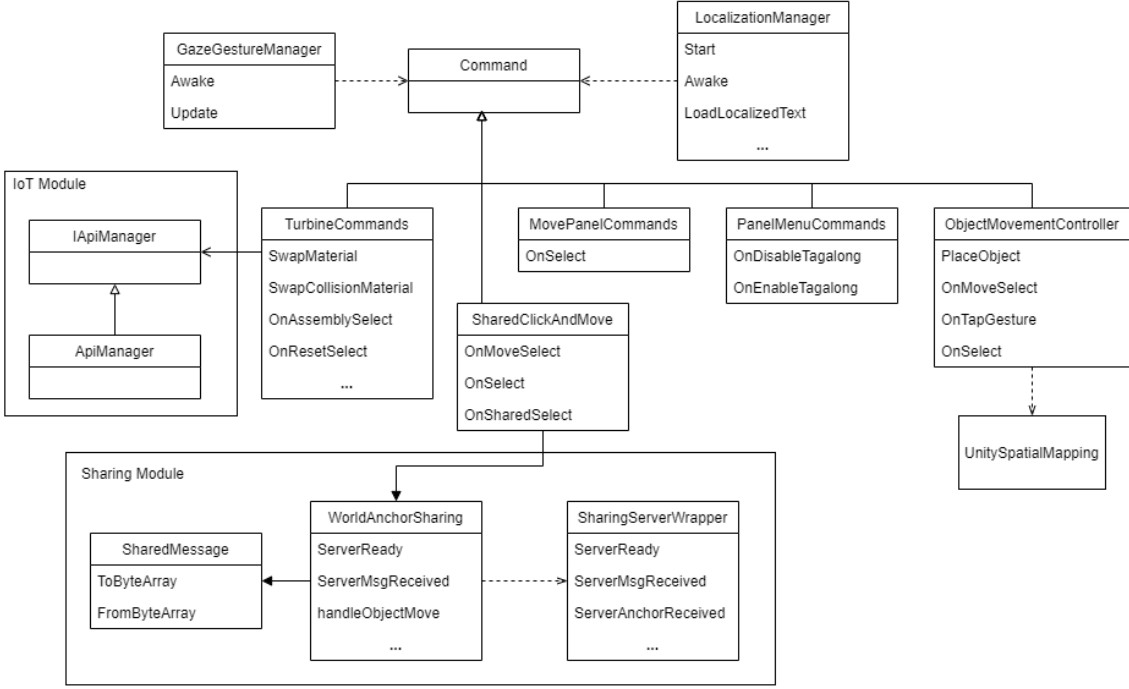

**Figure 6.** Simplified class diagram of the main components of the system.

The *TurbineCommands* script manages the animations and all the messages sent to the clutch from any other part of the scene (mainly from the buttons), while the Sharing module scripts are in charge of managing the shared experience subsystem. Moreover, there is an IoT module, which is in charge of managing the interactions with surrounding IoT networks. Finally, it is worth mentioning the *LocalizationManager* script, which handles the changes made on the user interface so that she/he can select from the different supported languages.

### 4.3.2. Implementation

As was previously mentioned in Section 4.1, the AR application was developed using Unity as the graphics engine. The 3D models were first created with Siemens NX 11 [59] and then were polished with Blender 2.83 [60] to make them lighter in terms of the number of polygons to be displayed and animated smoothly on AR devices.

The first step for the implementation was to convert and adapt the 3D engineering models to models suitable for AR representation. This step consists of converting the CAD models to a format that supports real-time rasterization. This process necessarily involves a loss of resolution when switching from a mathematical model to a model with finite points, and it is not a trivial process. On many occasions, it will require manual intervention or the use of Artificial Intelligence (AI) techniques to perform a more efficient conversion. Moreover, due to the AR device performance, it is necessary to make use of further optimization techniques such as polygon reduction or decreasing the resolution of textures in order to ensure a smooth experience when the application is used by the end user. As an example, Figure 7 shows the result of an automatic optimization procedure that reduces the number of vertices of a bolt of the clutch from 3267 to only 165. Microsoft provides a number of recommendations when it comes to implementing an AR application and indicates several problems that can arise when using CAD models for this purpose, such as flickering or Z-fighting [61].

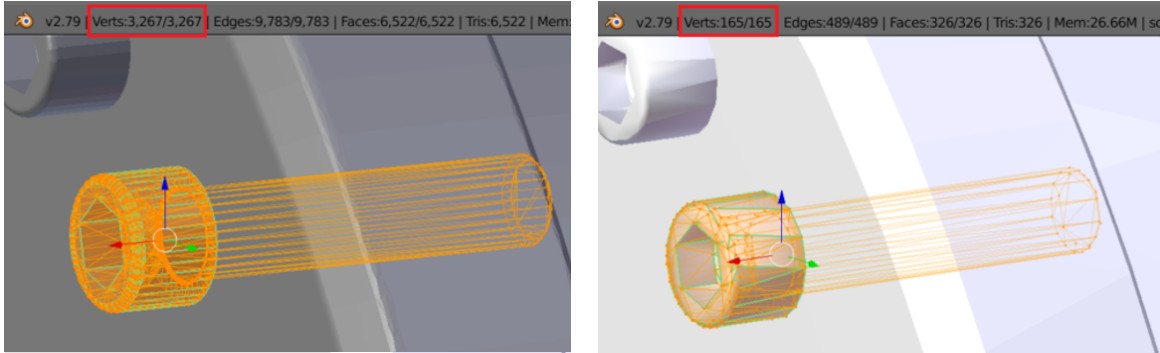

**Figure 7.** Bolt of the clutch before (**left**) and after (**right**) its geometry optimization (the red rectangle frames the number of vertices of the 3D model).

Once the 3D models were imported into the graphics engine (i.e., Unity), the clutch animations were created as specified by the documentation files provided by the shipyard workshop. An example of one of the animations is illustrated in Figure 8 (on the left), where all the clutch parts are visible after performing the disassembly animation. Then, the necessary scripts were implemented. Such scripts control the movement and the message reception of the rest of the components of the scene.

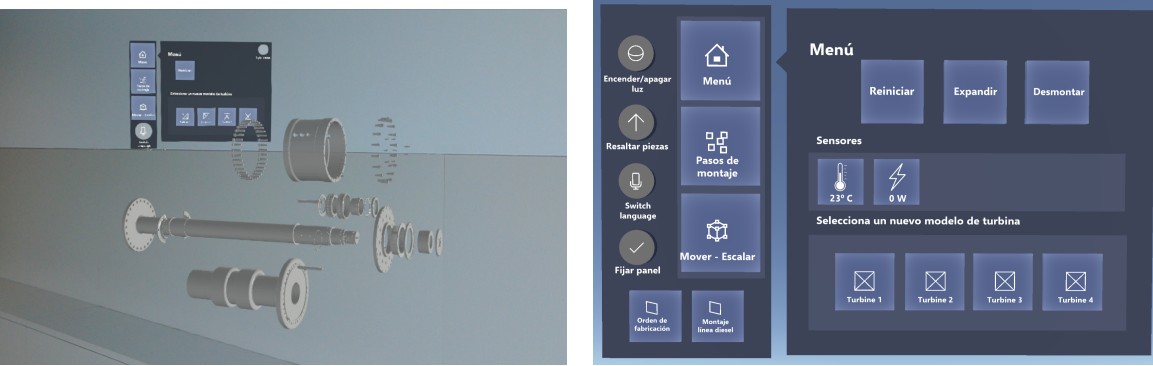

**Figure 8.** 3D model of the clutch after the disassembly animation (**left**) and a screenshot of the control panel (**right**).

The next step consists of building the user interface that includes a floating panel that follows the user and enables actions such as activating animations or visualizing information about the model. Figure 8, on the right, shows a close-up screenshot of the final version of the control panel. In addition, the documentation panels were added. Such panels show PDF documents associated with the manufacturing order and the assembly. In addition, as can be observed on the left of Figure 9, screenshots of the CAD drawings of the parts were also integrated into the panel, so when certain parts are selected, a panel with the associated documentation is shown.

The last step in the development consists of adding the functionality related to the collaborative framework, which was previously detailed in Section 4.2. Thanks to the protocol, the implemented system allows the different devices to share their location information on the different 3D elements in the environment, as well as the interactions that each user performs with the goal of enhancing the feeling of being immersed in the same virtual environment. As an example, Figure 9 illustrates two time instants when the operators made use of the shared experience and thus interacted simultaneously with the virtual hydraulic clutch.

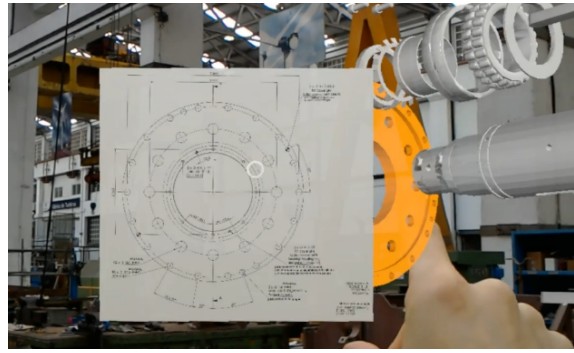 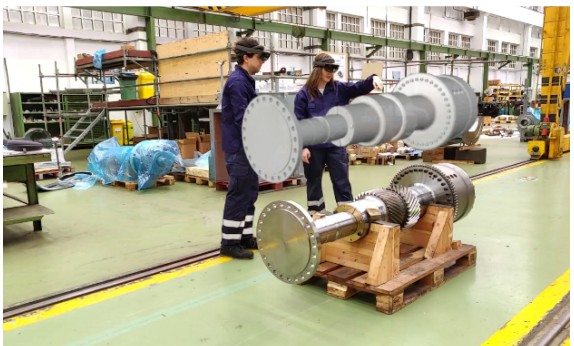

**Figure 9.** Operator consulting documentation (**left**) and two workers using the shared experience (**right**).

## 5. Experiments

### 5.1. Collaborative Framework Performance Tests

#### 5.1.1. Experimental Setup

Different experiments were carried out in order to determine the performance of the most innovative component of the proposed system: its collaborative framework. In such experiments, communication delay was measured when multiple AR nodes sent their position to all their neighbors continuously.

The following hardware was used for the tests:

- One pair of Microsoft HoloLens glasses.
- Router Linksys Wireless-G (2.4 GHz).
- Router Archer C3200 (5 GHz).
- Desktop computer.

In order to recreate a multi-user scenario with $n$ HoloLens glasses, a simulator was created using a Universal Windows Platform (UWP) application [62], which can be executed on a desktop computer. Such a kind of application is the most similar execution environment that can emulate in a desktop computer the functioning of HoloLens glasses, since the libraries developed for the sharing system can be executed natively.

The number of simultaneous devices can be selected on the HoloLens simulator. Once such a number is configured, the application starts to send the coordinates of the indicated amount of simulated glasses in parallel, using threads in a way that the network load for the actual glasses is comparable to the one that would exist if the same number of real glasses were sending packets over the network.

In order to evaluate the performance of the application, the main parameters that were measured were latency and packet lost. It is important to note that, in the type of application presented in this article, the delay of each individual packet is one of the most relevant parameters to provide a good user experience. In contrast to traditional real-time applications where network stability and jitter can have more impact than latency, this type of application has to be synchronized with the real world, meaning that when other users make a real gesture to interact with an object, the remote user has to perceive that the action is taking place in real time and synchronized with the real-world gesture. Therefore, it is more critical that each packet arrives as fast as possible instead of being stable over time.

#### 5.1.2. Regular Packet Communication Delay

The most common packet transmitted by the collaborative framework is really small (only 37 bytes), is sent via UDP every 200 ms, and contains the 3D location and orientation of an element, as well as some metadata that indicate which object has moved and how. The update period is configurable on the framework. It is set to 200 ms as the default value, as it provides a good

compromise between real-time movement and non-excessive network load. Such a value allows for rendering around 12 frames between each update when running above 60 frames per second, which allows for a good movement smoothing on the client.

With such packets, tests were performed for 10, 30, 40, 70, 90, and 110 concurrent users. The latency and the number of lost packets were tracked to determine how many users can be handled without harming the AR user experience. The tests were executed using two different wireless networks in order to evaluate how the wireless channel and protocol impacted the results: a 2.4 GHz network created with the Linksys router and a 5 GHz network operated by the TP-Link Archer router. The WiFi spectrum was checked before the tests to guarantee that the wireless routers operated in a WiFi channel with no overlapping networks.

Using such networks, latency was measured from the time the packet was sent to the network from the source device and until such a packet was decoded and processed by the target device. If the message contained by the packed included the new coordinates of an object, the latency was measured until the new coordinates were assigned to the target object. It is important to note that the measured latency did not include virtual object rendering time, since it depends on the number of polygons of the 3D objects shown in an AR scene and therefore is not related to the performance of the proposed collaborative framework.

In the previously described scenarios, the tests were performed several times and showed that the latency was very stable, requiring an average of 4.8 ms for the 2.4 GHz network and 3.3 ms when using the 5 GHz router. Such figures barely oscillate (only a fraction of millisecond) for the number of evaluated simultaneous users (between 10 and 110).

According to the International Telecommunication Union (ITU) definition of the Tactile Internet, a seamless video experience requires latencies of up 10 ms, but if a human expects fast reaction times, a 1-ms latency would be needed [63]. The obtained results are somewhat in the middle of both ITU's assessments, but, as analyzed in the next subsection, they do not depend so much on the collaborative framework, but on the underlying wireless communications infrastructure.

### 5.1.3. Interference Influence

Although the average latency barely varies for regular packet transmissions, as the number of concurrent users increases, more and more packets are dropped, the lag quickly becoming noticeable. This effect can be clearly observed when the nodes receive movement updates (i.e., the affected users see virtual objects "teleport" from one place to another). Such a behavior is justified in the view of Figure 10: the larger the number of concurrent devices, the larger the number of lost packets.

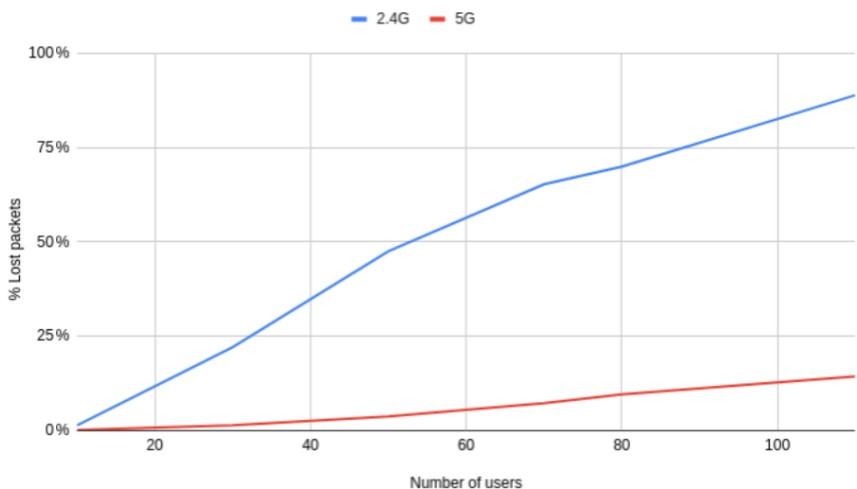

**Figure 10.** Linear representation of the number of packets lost over the number of users.

Despite the relevant number of packets lost in some cases, it is relevant to point out that, since UDP packets are sent every 200 ms, losing one is not a big deal. However, the percentage of lost packets can harm the user experience significantly in certain scenarios. For instance, for more than 30 users (an actually large number of users in applications that share the same AR experience in real time), in the case of the tested 2.4 GHz network, almost 25% of the packets are dropped, noticeably harming the user experience. However, it can be seen that network performance can be significantly improved by using the less crowded 5 GHz band and a slightly faster router: the number of lost packets drops to about 1%. With this type of network, the number of lost packets stays below 10% until there are more than 80 concurrent users.

### 5.1.4. Anchor Transmission Latency

Anchors are essential for the AR user experience, since they contain the environmental information necessary to determine the three-dimensional points where the virtual objects are. Therefore, it is interesting to measure anchor transmission latency.

It must be noted that such a latency is mostly determined by the size of the anchor generated by the Microsoft HoloLens framework, which depends on the complexity of the shapes of the real-world environment and on the amount of detail with which the AR devices have been able to map their surroundings. Due to these factors, the size of an anchor ranges from a few to several hundreds of megabytes. Thus, the transfer times of the anchors are proportional to their size.

In the performed tests, anchor transfer times ranged from 2 s for the smallest anchors to 30 s for the largest ones. These are large times, but it should be noted that the transfer of anchors is only necessary once when the AR application starts, in order to establish the same reference points for all the devices that share the AR experience. Once this initialization is performed, all the changes in coordinates are specified in relation to the initial reference points, so it is in general not necessary to transfer the anchor again.

### 5.2. Validation Tests

In order to validate the functioning of the developed solution, several lab tests were carried out and then repeated in Navantia's Turbine workshop. These tests allow for determining the practical viability of the system and for finding possible drawbacks or enhancements based on the operators' experience.

The laboratory tests were carried out at the university facilities with the aim of making a first validation of the implemented system, as shown in Figure 11. These tests were satisfactory, but detected a significant performance problem when the application's debugging system was activated. It should be noted that the debugging system must not be enabled for applications in production, but it is a problem worth verifying due to its significant impact on performance. This problem was corrected by deactivating the debugging also during the experiments.

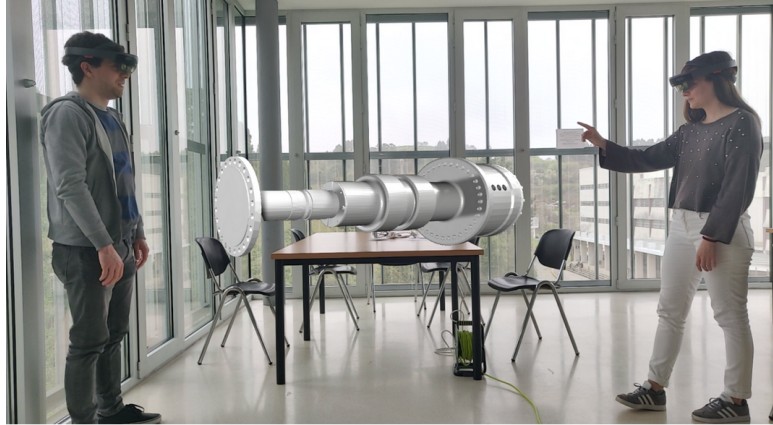

**Figure 11.** Developers testing the AR application at the university facilities.

After the laboratory tests, the same tests were carried out with the application, but this time in Navantia's Turbine workshop, as it is shown in Figure 12. The assembly of one of the hydraulic clutches in the turbine workshop was used to run the application and execute the assembly steps in parallel to the actual assembly of the parts. In this way, it was possible to determine the suitability of the system to the real environment and, additionally, to receive feedback from the operators about the application, which is analyzed in the next subsection.

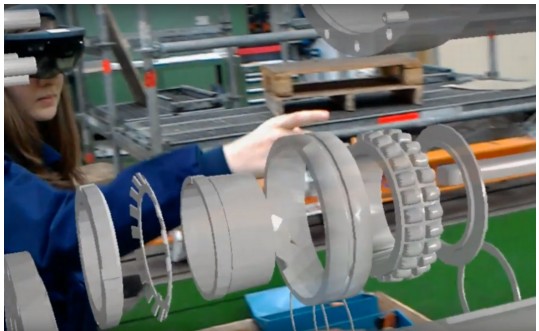 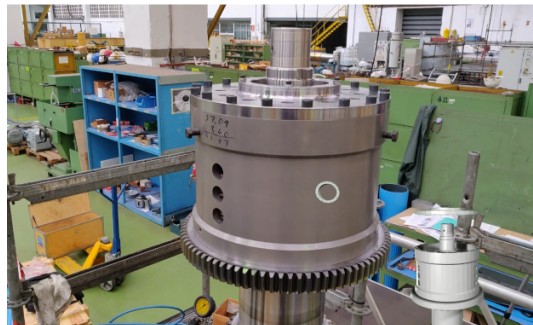

**Figure 12.** Interaction between two devices with a shared 3D element (**left**) and the assembly process at Navantia's Turbine workshop (**right**).

## 5.3. Operator Feedback

During the validation process, several operators in the turbine shop were taught how to use the AR application. The first impressions of the operators were rather surprising, as most of them had never tried any device with stereoscopic vision. In general, they said that they found it a very useful application, especially for the process prior to the assembly of the clutch, where this application would serve as a visual reinforcement to the documentation they currently have on paper. Of the ten male experienced operators (no female operators work in such a workshop) who tested the application, only one had trouble at first learning how to work with the application, since the gesture that has to be made to use the Microsoft HoloLens has to be quite precise. Apart from this small problem, the operators learned how to use the application perfectly. Several of them said that it was clear and very intuitive to use, besides the fact that the glasses are comfortable to use at the workshops through hands-free mode. A summary of the feedback given by the operators after testing the application can be found in Table 2.

**Table 2.** Summary of the feedback given by the operators after testing the application.

|  | Ease of Use | Intuitiveness | Application Usefulness | HoloLens Comfort |
|---|---|---|---|---|
| Excellent | 30% | 40% | 40% | 30% |
| Good | 60% | 50% | 40% | 70% |
| Can Be Improved | 10% | 10% | 20% | 20% |

Apart from this, the operators were also asked if they would like the application to have any more functionalities. The comments made by the operators were that they would like to see more detailed assembly steps and include the possibility of seeing sections of the parts (being able to cut them to see their interior in 3D). They also commented on the possibility of showing the points of collision or friction between the parts during the assembly process, as they noted that this is one of the most complicated steps of the assembly and in the blueprints is not fully understandable.

The general impression of the operators after testing the application was that they found the application to be very useful in their daily work, both for the assembly phase of the parts and in the previous steps, in order to learn how the parts are assembled without the need to have the actual clutch in front of them. This previous exercise would allow the assembly process to be much faster and safer, since operators would know in advance what difficulties they will encounter during the assembly process.

*5.4. Key Findings*

One of the most relevant difficulties that AR developers have to face is the fact that the used technologies are constantly changing and updating. This is due to the immaturity of the technologies, especially the Microsoft framework, which generates compatibility problems with the shared experience system and with Unity. It is also worth mentioning that multiple specific problems were found when using the Microsoft development tools, which sometimes produced unexpected results due to the issues related to such tools. However, in spite of this, it is fair to say that the development environment is one of the most advanced with respect to its competitors, and it is expected to continue improving.

Another related issue has to do with the creation and export of anchors. Anchors are created internally by the Microsoft framework, and their size depends on the knowledge of the environment that the HoloLens has in each moment and on the graphical complexity of the objects that are in the area. In some cases, the size of the anchors grows to such an extent that transmission over the network can take longer than expected, leading to failures when importing and exporting them. It is expected that, as the technology matures, the mentioned processes will become more reliable and less problematic.

As far as the functional part is concerned, it is worth mentioning that the developers' lack of knowledge of the domain and the manufacturing processes made some parts of the pilot development process difficult, especially when interpreting the manual and when designing animations. Because of this, future developers should consider carrying out a specific stage for defining in detail both the processes to be represented and the requirements and what is expected from the final application, emphasizing those parts that have greater difficulty or greater importance in the process. In this way, the team in charge of designing the application knows how to approach it.

Finally, based on feedback from the operators mentioned in Section 5.3, it is believed that a version of this application for mobile devices, such as tablets, could speed up the deployment of these tools in workshops. In addition, it would bring advantages such as facilitating the handling of traditional drawings in digital format. Furthermore, it would be possible to maintain the shown 3D part anchored to a surface in a real environment, so the operators could continue interacting with it in a very similar way to what they currently do. On the other hand, the use of smartphones or tablets would imply the loss of the stereoscopic vision provided by the glasses and the impossibility of handling the application hands-free.

*5.5. Next Challenges*

The developed AR system is useful in a real shipyard workshop, but future AR developers should consider the following challenges for their future work:

- As of writing, Microsoft is working on its own official framework to implement shared experiences. This could be an important step forward as it is thought that this official implementation would solve the compatibility problems detected during the development of the solution presented in this paper.

- Another important aspect that can be improved is the visualization of traditional documentation (or digital PDF files) through panels, which are shown as mere 2D objects that do not take advantage of all the potential of the used AR devices. In addition, documentation panels make it more difficult to navigate the information than in a computer or tablet. To tackle this issue, the documentation could be shown in a contextual way, associated with the parts that are being visualized in 3D and, whenever possible, using elements like arrows, gauges, tools (e.g., screwdrivers, drillers), or animations to enhance the information provided by the documentation.

- During the tests at Navantia's Turbine workshop, it was identified that the small details in the assembly steps were the most useful for the operators when using the developed AR application, since the possibility of watching the parts from different angles allows for highlighting specific details in a more effective way than a printed document. Therefore, future developers should consider adding more detail in each step of the assembly sequence, giving the possibility of moving back and forth in the animations.

- Due to the problems discussed in the previous Section 5.4 about the size of the anchors, it would be interesting to implement an error detection system during synchronization that would allow for restarting the synchronization and thus enabling automatically recovering from this type of malfunction.

- The option of adding speech recognition could be considered to facilitate the interaction with the application without the need for hand gestures, which would offer greater freedom when working. Nonetheless, developers should consider that speech recognition may be difficult or even impossible in noisy industrial environments.

## 6. Conclusions

IAR can play a significant role in the Industry 4.0 shipyard, but as of writing, there are only a few IAR training and assistance systems in the literature that consider explicitly shipbuilding requirements. In addition, current commercial systems are closed proprietary solutions that are difficult to integrate and adapt to current needs. In this article, a collaborative IAR system for training and assistance in assembly tasks is designed from scratch. After describing the proposed communications architecture, the implementation based on Microsoft HoloLens smart glasses is carefully detailed. The IAR system is evaluated to determine the performance of the collaborative framework proposed. According to the performed experiments, on average, less than 5 ms are required to exchange the framework's most frequent packets. Such packets are heavily affected by the interference that arises when multiple AR users communicate simultaneously. However, for the most typical shared experience (i.e., for less than 10 simultaneous AR users), interference is negligible. In addition, the system is also tested by several operators who provided feedback before a deployment in a real-world environment. As a conclusion, some best practices and lessons learned are outlined to guide future IAR developers.

**Author Contributions:** T.M.F.-C., P.F.-L., and M.V.-M. conceived and designed the experiments; A.V.-B. and O.B.-N. performed the experiments; P.F.-L. and T.M.F.-C. analyzed the results; A.V.-B., O.B.-N., T.M.F.-C., and P.F.-L. wrote the paper; T.M.F.-C., P.F.-L., and M.V.-M. revised the paper. All authors read and agreed to the published version of the manuscript.

**Funding:** This work was supported by the Plant Information and Augmented Reality research line of the Navantia-UDC Joint Research Unit. We wish to acknowledge the support received from the Centro de Investigación de Galicia "CITIC", funded by Xunta de Galicia and the European Union (European Regional Development Fund- Galicia 2014–2020 Program), by Grant ED431G 2019/01.

**Conflicts of Interest:** The authors declare no conflict of interest.

## Abbreviations

The following abbreviations are used in this manuscript:

AR　　Augmented Reality
CAD　　Computer-Aided Design
ERP　　Enterprise Resource Planning
HMD　　Head-Mounted Display
IAR　　Industrial Augmented Reality
IoT　　Internet of Things
IIoT　　Industrial Internet of Things
MES　　Manufacturing Execution System
MR　　Mixed Reality
PLM　　Product Life-cycle Management
UWP　　Universal Windows Platform
VR　　Virtual Reality

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
