# Peer review of "Creating Collaborative Augmented Reality Experiences for Industry 4.0 Training and Assistance Applications: Performance Evaluation in the Shipyard of the Futureâ€"

_applsci, doi:10.3390/app10249073_

Round 1
Reviewer 1 Report
The authors present a report on an augmented reality application for assistance in a shipyard. They show an interesting application in a niche that is otherwise not very well known. The main issue with this manuscript is the lack of evaluation on the effectiveness of the proposed system: In the goals of the research (section 3.1) the authors claimed that "The application seeks to reduce assembly times and to visually illustrate the contents of the assembly manuals, which in most shipyards are still only available for the operators on paper. Thus, the developed system is expected to optimize the processes and reduce the risks associated with errors during assembly." However, the manuscript falls short in showing whether these time/risk/error reductions were achieved or not. For example, the manuscript could gain a lot by showing the results of time assembly achieved by a control group not using the AR tool vs. a treatment group using it.
* Detailed comments on the text:
- In the abstract: latency of 30 seconds? By reading section 5.1.2, I thought it was about 3 ms, which one is correct?
- The Industry 4.0 paradigm may not be known by all readers, a more detailed description of it would be useful in the introduction (and or some references for the interested reader).
- "associated with every relevant,": missing noun?
- "Sounds and sound effects that help the user to recognize and reaffirm different actions, such as the click or the start and end of the animations." I think the authors mean "earcones." See: [1] M. Cohen and J. Villegas, “Applications of audio augmented reality. Wearware, everyware, anyware, and awareware,” in Fundamentals of Wearable Computers and Augmented Reality, ch. 13, pp. 309–329, CRC Press, 2nd ed., July 2015.
- One of the most important contributions of this work was self-identified by the authors to be the collaborative protocol. Regarding this, there are several things that could be improved:
-- What other methods were evaluated before selecting the implemented one?
-- There are protocols already implemented (such as http://opensoundcontrol.org/introduction-osc) that could deal with these requirements. Why was the implemented method chosen?
-- How are conflicts resolved with the implemented protocol? E.g., two users simultaneously move the model in opposite directions, how is this being solved in your solution?
- Please add the description of the solid and dashed line to figure 2 (I know it's already in the text).
- Please add references to Unity, Siemens NX, Blender, and the versions used.
- Reference needed for "Microsoft provides a number of recommendations when it comes to implementing an AR application and indicates several problems that can arise when using CAD models for this purpose, such as flickering or Z-fighting."
- Reference needed for "Universal Windows Platform (UWP) application"
- How was the period of 200 ms determined? shouldn't this be dependent on the task?
- The authors made a detailed analysis of the delays of the networks, what about the jitter? in many cases, jitter is more problematic than delays.
- Were the users informed of the progress on the transfer of the anchors?
- "These tests were satisfactory, but detected a significant performance problem when the application’s debugging system was activated. This problem was corrected by deactivating the debugging." Why is this important? From a developer's perspective I understand that debugging is needed, but in production, why would this feature be active?, and why is it being noted in the manuscript?
- "During the validation process, several operators in the turbine shop" how many? how much experience they had? how many male/female. This section deserves a more complete description. In fact, this should be an experiment with a control and experimental group as mentioned previously.
- There seems to be a lot of work after the proceeding publication related to this article. But, besides the note above the abstract, the authors should add a paragraph in the introduction explaining how these two publications differ.
- One of key findings in the study is that because of how Microsoft framework creates anchors, sometimes these can be large enough to cause problems. Is it possible to integrate Multi-resolution models such as those discussed in:
https://openaccess.thecvf.com/content_cvpr_2016/papers/Blaha_Large-Scale_Semantic_3D_CVPR_2016_paper.pdf
and
https://link.springer.com/chapter/10.1007/978-3-642-78114-8_29
?
* quibbles and typos:
- "previously mentioned works ": previously mentioned work.
- "Camera: it is the Unity camera used to render the objects in the scene. At the beginning of the execution it must be located in the XYZ (0,0,0) coordinates." Where is the camera looking at originally?
- Position comprises location (as in x,y,z coordinates) and orientation (rotations around those axes). So the text could be reviewed to use the correct term when necessary.
Author Response
Dear Sir/Madam,
The authors would like to thank the reviewer for his/her valuable comments, which have certainly helped us to improve the manuscript. Please find attached our detailed responses to the comments. In order to ease the labor of the reviewers we have colored in red the differences with the previous version of the article.
Best regards,
The authors.

Reviewer 2 Report
This paper describes a collaborative augmented reality system, with special focus on (1) the packet communications delay and anchor transmission latency and (2) the validation from workshop operators. The paper general quality is good:
1- The structure is clear, and the abstract, objectives and conclusions are consistent.
2- The writing is clear, with good English quality.
3- The results are relevant,
Things to be improved:
1- The paper tittle could put more emphasis on the results, namely the communications performance)
2- The state of the art should be improved, including relevant papers in fields that share a lot with the shipbuilding sector, like construction: Examples:
https://www.researchgate.net/publication/333931327_Collaborative_Welding_System_using_BIM_for_Robotic_Reprogramming_and_Spatial_Augmented_Reality/figures?lo=1
https://www.researchgate.net/publication/254040166_A_Study_on_Construction_Defect_Management_Using_Augmented_Reality_Technology
3- The results differ significantly in quality, between the communication analysis and the validation by the shopfloor worker. Although it is difficult to collect feedback from the end-user experience, a more formal approach, questionnaires for example, would significantly improve the quality of the paper.
Author Response

(The authors gave the same response as above.)

Reviewer 3 Report
The reviewer found this paper well written and have only one question regarding human studies:
Chapter 5.3 – if there were any survey – it would be glad to see some recap in the table format. Also, how many people were being experimented with the AR application?
Author Response

(The authors gave the same response as above.)

Round 2
Reviewer 1 Report
The authors have addressed my concerns and I think this article is ready for publication.